# Breath Measurement Method for Synchronized Reproduction of Biological Tones in an Augmented Reality Auscultation Training System

**DOI:** 10.3390/s24051626

**Published:** 2024-03-01

**Authors:** Yukiko Kono, Keiichiro Miura, Hajime Kasai, Shoichi Ito, Mayumi Asahina, Masahiro Tanabe, Yukihiro Nomura, Toshiya Nakaguchi

**Affiliations:** 1Department of Medical Engineering, Graduate School of Science and Engineering, Chiba University, 1-33 Yayoicho, Inage-ku, Chiba-shi 263-8522, Chiba, Japan; 2Department of Cardiovascular Medicine, Graduate School of Medicine, Chiba University, 1-8-1 Inohana, Chuo-ku, Chiba-shi 260-8670, Chiba, Japan; markone@hospital.chiba-u.jp; 3Department of Respirology, Graduate School of Medicine, Chiba University, 1-8-1 Inohana, Chuo-ku, Chiba-shi 260-8670, Chiba, Japan; daikasai@chiba-u.jp; 4Department of Medical Education, Graduate School of Medicine, Chiba University, 1-8-1 Inohana, Chuo-ku, Chiba-shi 260-8670, Chiba, Japan; sito@faculty.chiba-u.jp; 5Chiba University Hospital, 1-8-1 Inohana, Chuo-ku, Chiba-shi 260-8677, Chiba, Japan; asahi-to-yuuhi@faculty.chiba-u.jp; 6Chiba University, 1-33 Yayoicho, Inage-ku, Chiba-shi 263-8522, Chiba, Japan; hirot@faculty.chiba-u.jp; 7Center for Frontier Medical Engineering, Chiba University, 1-33 Yayoicho, Inage-ku, Chiba-shi 263-8522, Chiba, Japan; ynomura@chiba-u.jp

**Keywords:** wearable device, thoracic variation, inertial sensor, breathing parameters, digital filtering, auscultation training simulation

## Abstract

An educational augmented reality auscultation system (EARS) is proposed to enhance the reality of auscultation training using a simulated patient. The conventional EARS cannot accurately reproduce breath sounds according to the breathing of a simulated patient because the system instructs the breathing rhythm. In this study, we propose breath measurement methods that can be integrated into the chest piece of a stethoscope. We investigate methods using the thoracic variations and frequency characteristics of breath sounds. An accelerometer, a magnetic sensor, a gyro sensor, a pressure sensor, and a microphone were selected as the sensors. For measurement with the magnetic sensor, we proposed a method by detecting the breathing waveform in terms of changes in the magnetic field accompanying the surface deformation of the stethoscope based on thoracic variations using a magnet. During breath sound measurement, the frequency spectra of the breath sounds acquired by the built-in microphone were calculated. The breathing waveforms were obtained from the difference in characteristics between the breath sounds during exhalation and inhalation. The result showed the average value of the correlation coefficient with the reference value reached 0.45, indicating the effectiveness of this method as a breath measurement method. And the evaluations suggest more accurate breathing waveforms can be obtained by selecting the measurement method according to breathing method and measurement point.

## 1. Introduction

Auscultation of breath and heart sounds constitutes one of the clinical skills in physical examination; auscultation is a medical technique used to listen to lung airflow and heart sounds to obtain biological information noninvasively and quickly [1]. Since these sounds vary greatly depending on the auscultation position and presence of diseases, auscultation requires sufficient training for clinical practice [2]. Recent studies have demonstrated a decline in auscultation skills in clinical practice, and there is growing interest among practitioners to improve auscultation training [3,4,5,6]. Against this backdrop, current medical educational institutions conduct the objective structured clinical examination (OSCE) before students advance to clinical practice [7,8]. This examination aims to evaluate the proficiency of clinical skills such as “basic examination skills” and “communication skills with patients”, which are difficult to evaluate by paper tests.

In current auscultation training and OSCE, medical interviews and auscultation techniques are performed on a simulated or standardized patient (SP) who reproduces the physical findings of the disease. Since the SP is healthy, the numbers of breath and heart sounds that can be reproduced are limited. Hence, many medical education institutions use a method that combines medical interviews with auscultation procedures on a mannequin-type auscultation simulator [9,10]. Simulators have been shown to enhance physical examination skills of learners, and learners have also been shown to value simulation-based teaching very highly [11,12]. However, this method requires the trainees to turn around and face the simulator when shifting from the medical interview to the auscultation procedure, which may reduce the reality and training efficiency. In addition, the simulator is very expensive and requires ample space for storage and use, making it difficult to implement in facilities that are not large.

To overcome these problems, Nakaguchi et al. [13] proposed a virtual auscultation simulator for auscultation training on an SP and showed its effectiveness in reproducing training by comparison with a mannequin-type simulator. In addition, Sekiguchi et al. [14,15] proposed the educational augmented reality auscultation system (EARS), an augmented-reality-based auscultation training system, using deep learning; this system reproduces the breath and heart sounds of various diseases on a healthy SP and enables medical interviews as well as auscultation procedures to be performed on the SP. The trainee performs auscultation on the SP using a special stethoscope, similar to a regular medical examination; this stethoscope is divided into the chest piece and camera unit parts (Figure 1). The chest piece part has a built-in contact sensor that determines whether the patient is being auscultated by the presence or absence of contact with the body. The camera in the camera unit part acquires the positions of the chest piece during auscultation along with those of the SP’s head, shoulders, and hips during non-auscultation to calculate the auscultation position on the SP’s body based on the position information. EARS also plays the sounds of selected diseases according to the calculated auscultation positions.

However, when using the conventional system, the SP must breathe according to the timing shown on the breath indicator mounted on the camera unit of the stethoscope to match the rhythm of thoracic variations and breath sounds played. This burdens the SP heavily and makes it impossible to respond to deep breaths and breath holding. To solve this, it is necessary to measure the breathing of the SP during auscultation training.

Breath measurement methods can be broadly classified into contact and noncontact types. As a contact-type breath measurement method, a belt with an inertial or a piezoelectric sensor is attached to the chest [16,17]. This method measures the thoracic variations caused by breathing using sensors built into the belt. Another contact-type method involves attaching a microphone to the chest and investigating the exhalation and inhalation sounds of breathing to estimate respiration [18]. The disadvantage of such contact-type methods is the increased preparation time required for attaching the sensor. As a noncontact breath measurement method, the movements of the thorax are acquired using a depth camera and an ultrasonic proximity sensor [19,20]. This method involves monitoring breathing motions by measuring the distance to the thorax from the camera or sensor. However, the main disadvantages of this method are that the accuracy of acquisition depends on the positional relationship between the SP and camera or sensor, which requires time to setup, and that the accuracies of the breath measurements depend on the movements of the SP and wrinkles in the clothing accompanying the movements. There is also a method to measure breathing based on temperature changes around the nasal cavity of the SP using a thermal imaging camera [21,22]; however, this method has the disadvantage that it can only be applied when the SP is facing the front of the camera. These noncontact types have the disadvantage of requiring unique cameras, and when mounted, the EARS device becomes expensive.

In the present study, we propose a method to measure breathing by contact during auscultation training to realize the reproduction of breath sounds based on the breathing of the SP in EARS. To reduce the time and effort required to attach the sensor to the SP, which is a drawback of contact-type breath measurement, a method that can be integrated into the chest piece of a stethoscope is investigated. We propose a new principle for real-time acquisition of breathing waveforms, then evaluate the method and discuss how it can be used in a system. The remainder of this manuscript is organized as follows. Section 2 describes the proposed breath measurement methods. Section 3 presents the experimental setup and measurement accuracy of each method. Section 4 presents analyses of the measurement accuracies and discusses their prospects. Finally, Section 5 presents the conclusions of this study.

## 2. Materials and Methods

In this study, we investigate two methods for measuring breathing using thoracic variations and breathing sounds. For measurement devices, we developed ones that can be measured using only a chest piece and that are close to the size and weight of an actual chest piece to maintain the realism of the actual auscultation procedure.

### 2.1. Breath Measurement Using Thoracic Variations

Humans breathe by taking oxygen into and expelling carbon dioxide from the alveoli. Breathing is mainly achieved by the diaphragm and external intercostal muscles, which expand and contract the lungs. The movements of the diaphragm and external intercostal muscles cause thoracic variations [23]. During inhalation, the diaphragm and external intercostal muscles contract. When the diaphragm contracts and descends, the contraction of the external intercostal muscles expands the thorax back and forth so that the lungs expand, resulting in inhalation. During exhalation, the diaphragm and external intercostal muscles relax. The diaphragm returns to its original position, while the thorax narrows and lungs contract by elastic contraction. In this study, we propose a method to measure breathing motions using such movements of the thorax.

#### 2.1.1. Thoracic Variability Measurement Device Design and Data Collection

A stethoscope-type measurement device was fabricated using a 3D printer to measure the thoracic variations; this device is shown in Figure 2. With this device, breath measurements can be performed by placing the device on the SP’s chest as in the actual auscultation technique. The side and bell parts of the chest piece are made of polylactic acid resin, and the diaphragm surface is made of a deformable thermoplastic polyurethane resin. A pressure sensor FSR402 (Interlink Electronics Inc., Camarillo, CA, USA), an indenter, and a magnet are placed on the diaphragm surface and are connected to a breadboard and a 9-axis inertial sensor Adafruit Precision NXP 9-DOF Breakout Board—FXOS8700 + FXAS21002 (Adafruit Industries, New York City, NY, USA). The 9-axis inertial sensor includes an accelerometer, a gyro sensor, and a magnetic sensor. The characteristics of these sensors are shown in Table 1 and Table 2. Arduino UNO (Arduino Holding, Ivrea, Italy) was used to interface the sensors with the PC.

#### 2.1.2. Measurement by Parallel Shift Movement

An accelerometer was used to measure the movements of the thorax in terms of the amount of parallel shift. In addition, a measurement mechanism using a magnetic sensor and a magnet was developed to measure the amount of movement of the stethoscope surface caused by the thoracic movements. In this mechanism, a small permanent magnet is placed on the stethoscope surface, which is the contact surface, and the magnetic sensor measures the slight variations in the magnet due to deformation of the surface from changes in the magnetic field. The measurement principle is shown in Figure 3. A lowpass filter was then applied to each sensor output and processed high-frequency noise by the circuitry and unstable signal components.

#### 2.1.3. Measurement by Angular Displacement

Accelerometer and gyro sensors were used to measure the movements of the thorax as rotational motions. The sensor values were corrected using a lowpass filter and a Kalman filter.

#### 2.1.4. Pressure Measurement

A pressure sensor was used to measure the pressure on the stethoscope’s chest piece in contact with the measurement surface caused by the back-and-forth movements of the thorax due to breathing motions. The pressure sensor used here is the one described in Section 2.1.1. Here, an indenter for noise reduction was created with a 3D printer and placed on the back of the pressure sensor, as shown in Figure 2.

### 2.2. Breath Measurements Using Breath Sounds

The overall sounds produced by breathing are called lung sounds, which are classified into two categories as breath sounds (airflow sounds during ventilation of the airways and alveoli) and sub-noise (abnormal sounds generated in pathological conditions) [24]. In this study, we focus mainly on the breathing sounds. Three types of breath sounds can be heard under normal conditions: bronchial, bronchoalveolar, and alveolar. Bronchial breath sounds are louder than those heard from the other parts of the body and are particularly louder during exhalation than inhalation, with a clear pause between inhalation and exhalation. The frequency components of the sound are around 600 Hz for inhalation and 400 Hz for exhalation. Bronchoalveolar breath sounds can be heard more clearly than alveolar breath sounds during both inhalation and exhalation. The loudness of the breath sounds is the same for inhalation and exhalation, or slightly louder for inhalation, and the frequency components are the same as those for bronchial breath sounds, which are around 600 Hz for inhalation and around 400 Hz for exhalation. Alveolar breath sounds are clearly audible upon inhalation, but the exhalation sounds are quiet and difficult to hear. The ratio of inhalation to exhalation time is approximately 1:2. The frequency components of the inhalation sounds are around 400 Hz and those of the exhalation sounds are around 200 Hz [25,26]. We propose a method to measure the breathing phase using these differences between the exhalation and inhalation sounds.

#### 2.2.1. Related Research

Li et al. [27] proposed a breath sound analysis algorithm for real-time whistle sound detection; this algorithm extracts the characteristics of breath sounds in both time and frequency domains and classifies normal breath sounds and whistle sounds by taking advantage of the frequency range of the whistle sounds (250–800 Hz).

The steps in this method are as follows:The breath sounds are bandpass filtered in a passband of 150–1000 Hz. This step is used to remove heart sounds and signals from muscle interference [28].The data are subjected to a Fourier transform using a Hanning window (short-time Fourier transform). Spectral integration (SI) and normalized spectral integration (NSI) are then calculated to obtain the characteristics of the breath sounds in the frequency domain. The NSI is the ratio of the value of each SI when the sum of the calculated SI is set to 1.The NSI values are used to classify normal breath sounds from whistle sounds using Fisher linear discriminant analysis.

#### 2.2.2. Proposed Methodology for this Study

In this study, we propose a breathing waveform acquisition method using the ratio of SIs as well as the SI and NSI described above, as outlined in Figure 4. First, the breath sounds are denoised using a moving average filter of size 10 samples, and a short-time Fourier transform is then performed using a Hanning window. The window width is set to 4096 points and overlapped so that the frameshift is 50 ms. The ratio of spectral integrals is then calculated from the obtained spectra. As shown in Figure 5, in this study, the ratio of SIs in the range of 500–800 Hz out of those in the range of 100–1000 Hz, which is the frequency range of breath sounds, is calculated as the values of the breathing waveform. The calculated values are then multiplied with a moving average filter of size 10 samples and used as the final breathing waveform acquired by this method.

#### 2.2.3. Breath Sound Sampling Device Design and Data Collection

A stethoscope-type breath sound sampling device with a microphone inserted into the stethoscope tube was fabricated to measure the breath sounds. The fabricated device is shown in Figure 6. The stethoscope used was a Flare Phonet No. 137II (KENZMEDICO CO. LTD., Saitama, Japan), and the microphone used was ECM-PC60 (Sony Group Corporation, Tokyo, Japan). The microphone characteristics are shown in Table 3; the microphone was connected to a laptop computer via a conversion adapter, and its sampling rate was 44.1 kHz.

The chest piece has two types of listening surfaces, namely the diaphragm and bell surfaces. In this experiment, the diaphragm surface was used to measure the breath sounds.

## 3. Results

### 3.1. Experimental Procedures

A total of 14 subjects (7 females and 7 males) in their early 20s without any health problems were recruited for the study. Since auscultation is performed on healthy subjects in the EARS system, we selected healthy subjects for this breath measurement. Informed consent was obtained from all subjects involved in the study. Based on actual EARS use, measurements were obtained while the subjects were wearing clothing. Five measurement points (four in the front and one in the back) were selected from the basic auscultation positions for the breath sounds (Figure 7). In this study, measurements were taken only on the right side of the body, which is less affected by the heartbeat. To reproduce an actual medical interview, each subject was instructed to take either normal or deep breaths for each of the measurements. The amount of movement of the chest and abdomen varies depending on the breathing method, which may result in differences in accuracy depending on the measurement point and method [29]. Three 20-s measurements were acquired per subject at each measurement position and for each breathing method. All subjects wore the same cotton T-shirt to minimize any influence from clothing. The experiments were conducted in a sitting position to simulate a medical interview.

The signal from the Go Direct Respiration Belt (Vernier Science Education, Beaverton, OR, USA) was used as the reference for the breathing waveform. The specifications of the sensor are shown in Table 4. The sensor was placed above the subject’s dovetail (blue circle in Figure 7). From now on, the signals obtained with this respiration measurement belt are referred to as the biometric sensor signals.

### 3.2. Evaluation Method

The normalized cross-correlation function was used as the evaluation index. The equation of the cross-correlation R for two time-series waveforms (x,y) and number of data N is shown below.
(1)$$R(∆t)=\frac{1}{N-∆t}\sum_{t=1}^{N-∆t}x\left(t\right)y(t+∆t)$$

After normalizing the reference biometric sensor signal and breathing waveform signals from the data acquired by the thoracic variation measurement device and breath sound sampling device, cross-correlation analysis was performed to obtain the correlation coefficient of the waveform with the reference signal. Then, an analysis of variance was performed on the obtained correlation coefficients to test the differences in accuracies of acquiring breathing waveforms by different measurement methods. For the analysis of variance, anovakun ver4.8.6—a script for analysis of variance on the statistical software R—was used. The version of the statistical software R used was 4.0.0.

### 3.3. Experimental Results

Figure 8 shows the breathing waveforms obtained for each measurement method, and Figure 9 compares the cross-correlation coefficients among the measurement methods. It was confirmed that the average value of the correlation coefficient for each measurement method was more than 0.4, which is considered to be correlated. However, the final target value for the correlation coefficient is 0.7, which is considered a strong correlation, so the results indicate that we have not yet reached that level.

A three-factor analysis of variance for mixed designs was conducted for the number of interrelationships based on the breathing method (within-participant items: 2 levels), measurement position (within-participant items: 5 levels), and measurement technique (within-participant items: 5 levels). These results are shown in Table 5; the results show that the second-order interactions were insignificant. However, significant differences were obtained in the first-order interactions between the breathing and measurement method factors and also in the first-order interactions between the measurement position and measurement method factors. Table 6 shows the means and standard deviations for the corresponding breathing method and measurement technique factors. Table 7 shows the means and standard deviations for the measurement position and technique factors. Table 8 shows the result of simple main effects for each interaction, and Table 9 shows the results of Bonferroni’s multiple comparisons for those factors for which the main effects were significant. In the table, SS represents the sum of squares, Df represents degrees of freedom, and MS means mean squares.

## 4. Discussion

The breathing method and measurement position are factors that change the thoracic variations and types of breath sounds. The analysis of variance showed that differences in the accuracy of acquiring breathing waveforms between the measurement methods based on thoracic variations were caused by differences in the breathing method and measurement position.

In terms of the breathing method, deep breaths tended to be abdominal breathing, compared to normal breaths, and we considered a significant difference in the accuracy of breathing waveform acquisition [29]. Breathing can be broadly classified into two types: as costal and abdominal breathing. Costal breathing is a method in which mainly the thorax moves, and the muscles that work during inhalation are the external intercostals and respiratory accessory muscles, while the muscles working during exhalation are the internal intercostals. During inhalation, the external intercostals and several respiratory accessory muscles are strongly contracted to raise the thorax anteriorly and upward to assist inhalation. During exhalation, the internal intercostals contract to contract the thorax and assist in exhalation. Abdominal breathing is a method in which the movement of the diaphragm is the main component. The diaphragm is the muscle that works during inhalation, and the abdominal muscles work during exhalation. During inhalation, the diaphragm contracts strongly, expanding the thorax in a large vertical direction. During inhalation, the diaphragm contracts strongly, and the thorax expands vertically. Thus, the differences in the movements of the thorax depending on the breathing method are considered to be the reason for the significant differences among the three methods for detecting the movements of the thorax. The results of the analysis of variance showed that the interactions between the breathing and measurement methods were significant and that these effects were considerable. Considering this, we selected the pressure sensor for normal breath and magnetic sensor for deep breath as the measurement methods and analyzed the results again, as shown in Table 10. The accuracy of breath measurement was improved compared to the case where no method was selected, suggesting that the selection of the measurement method according to the breathing method is effective.

The correlation coefficients were significantly different at the measurement points due to differences in thoracic variations by position and breath sounds. The correlation coefficients for the magnetic sensor measurements were significantly higher at measurement points (4) and (2) than at the other points. This is likely because the measurement points are closer to the upper thorax and abdomen than the other points, and thorax variation is more significant in these points. In addition, the correlation coefficients were significantly higher at the measurement point (2) than at the other measurement points for the accelerometer and gyro sensor measurements. This is because the measurement point (2) is close to the top of the thorax, and the stethoscope is placed at an angle to the body compared to other points, so the thoracic variations are easily expressed as angular displacements. Thus, the accuracy of breathing waveform acquisition may change depending on the difference in thoracic variation depending on position. These results are used to select the most appropriate breath measurement method according to the measurement position. The correlation coefficient for breath sounds was lower near measurement point (4) than at other points, but we consider that this was due to the fact that it was in the peripheral region of the bronchi. This is because the breath sounds are generated only in the bronchi from the oral cavity to the seventh to ninth branches, which are turbulent regions, and the breath sounds therefore tend to become lower than those in other regions as they get closer to the end region of the bronchi, being easily affected by noise such as clothing. In addition, in the peripheral region of the bronchi, the percentage of alveolar sounds is higher than bronchial sounds in the collected breath sounds. Therefore, as described in Section 2.2, the frequency range of exhalation and inhalation sounds tended to be lower than that of the bronchial sounds, making it difficult to obtain breathing waveforms in the frequency range selected for this experiment. In this work, the ratio of SI in the range of 500–800 Hz was acquired as the breathing waveform. However, if the ratio of SI in the range of 300–600 Hz was acquired as the breathing waveforms for the alveolar sounds, the results would be as shown in Table 11. Therefore, acquiring more accurate breathing waveforms for breath sounds is possible by changing the preprocessing according to the position as well as the frequency band of the exhalation–inhalation discrimination.

One limitation of this research is that the quality of circuit design and component mounting needs to be improved because this research aims to confirm the principle of a new respiration sensing method. Collaborating with hardware design experts for future practical use is necessary.

## 5. Conclusions

In this study, we propose breath measurement methods for synchronized reproduction of breath sounds in a stethoscope-based training system. Two methods are proposed here, namely those based on thoracic variations and breath sounds. For the breath measurement method using thoracic variations, a pressure sensor, an accelerometer, a gyro sensor, and a magnetic sensor were selected for the measurements, and a stethoscope-type measurement device was created using a 3D printer. For the breath measurement method using breath sounds, a measurement device using a stethoscope and a microphone was created, and a method for acquiring breathing waveforms was proposed to discriminate exhalation and inhalation sounds using spectral integration and conversion to waveforms.

The evaluation experiments showed correlations with the reference values of the breathing waveform, but strong correlations could not be obtained. For each method, the value of the correlation coefficient was higher when the measurement method with the highest correlation coefficient was selected, suggesting that the breath measurement performance can be improved by selecting an appropriate method.

The contribution of this paper is that we devised a new principle that enables an effective and real-time acquisition of breathing, under the constraint that it can be built into the chest piece, and evaluated its effectiveness. In the future, we intend to systematically study selection of the measurement method according to measurement conditions. We also aim to examine the optimal method of dealing with noise and signal preprocessing, which this study could not examine to obtain a stronger correlation with reference values. Finally, we would like to consider implementation of a breath sound synchronization reproduction method for auscultation training systems.

## Figures and Tables

**Figure 1 sensors-24-01626-f001:**
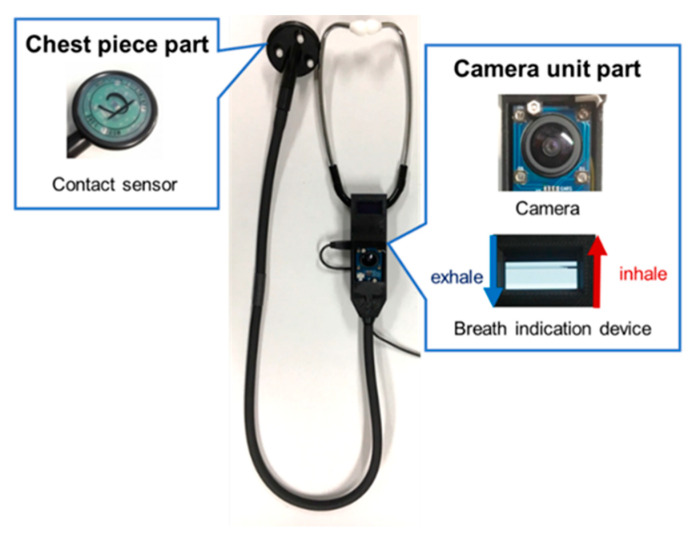
Conventional stethoscope used in EARS.

**Figure 2 sensors-24-01626-f002:**
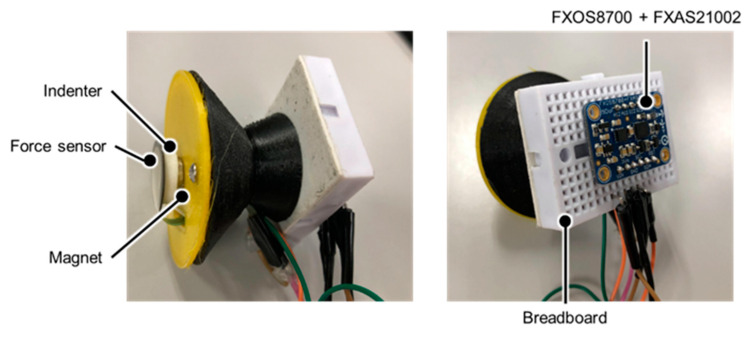
Thoracic variability measurement device.

**Figure 3 sensors-24-01626-f003:**
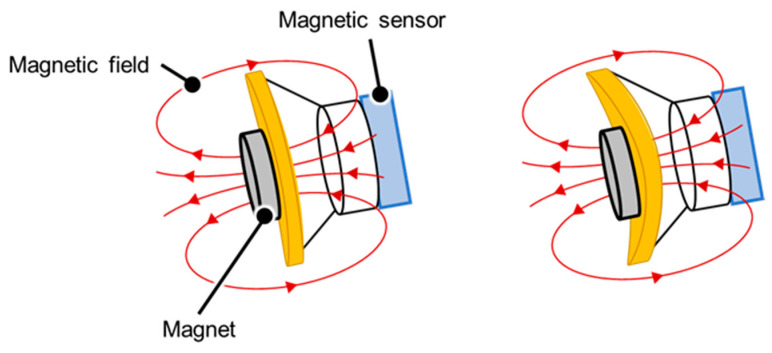
Principle of measurement by the magnetic sensor. When the surface of the stethoscope is indented due to thoracic variations, the magnetic field emitted by the magnet shifts and changes the value of the magnetic sensor.

**Figure 4 sensors-24-01626-f004:**
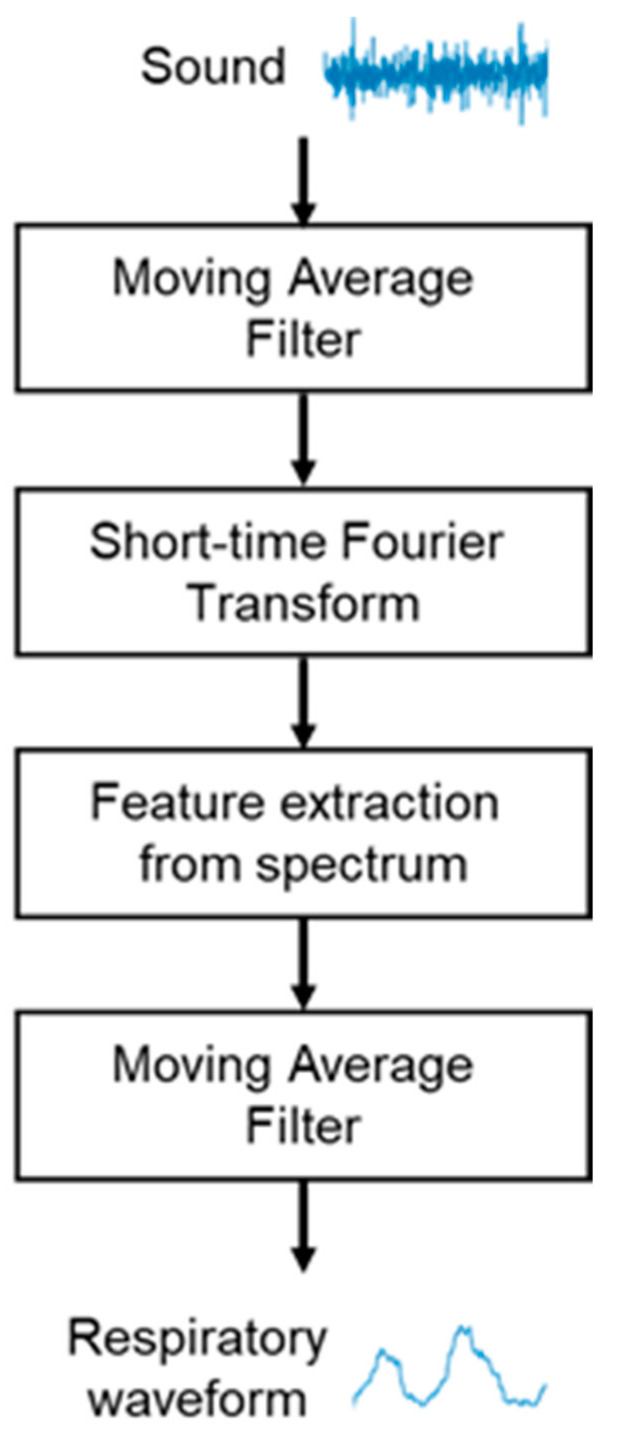
Flowchart for breathing waveform acquisition using breath sounds.

**Figure 5 sensors-24-01626-f005:**
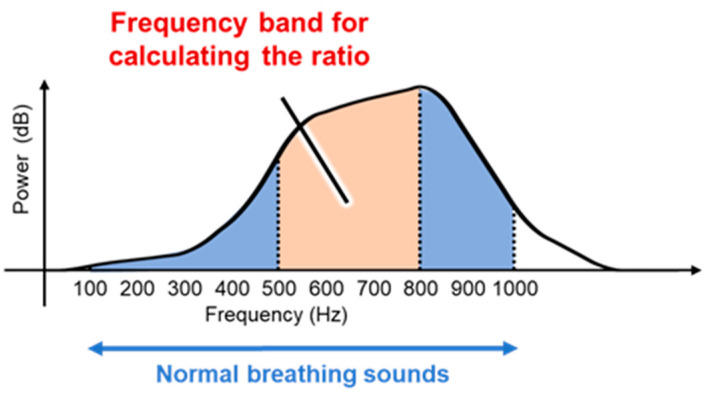
Spectral integral ratio range of breath sounds calculated using the proposed method.

**Figure 6 sensors-24-01626-f006:**
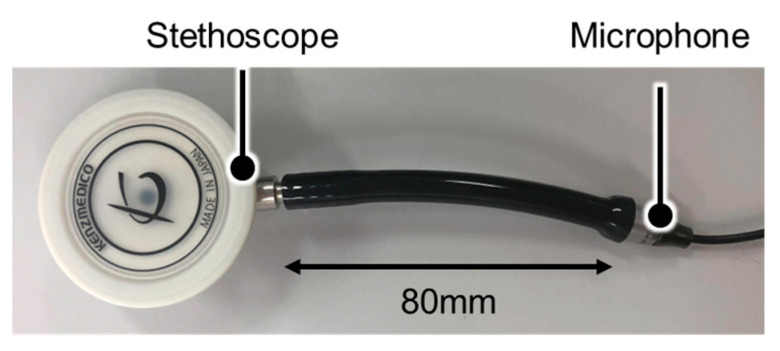
Breath sounds sampling device.

**Figure 7 sensors-24-01626-f007:**
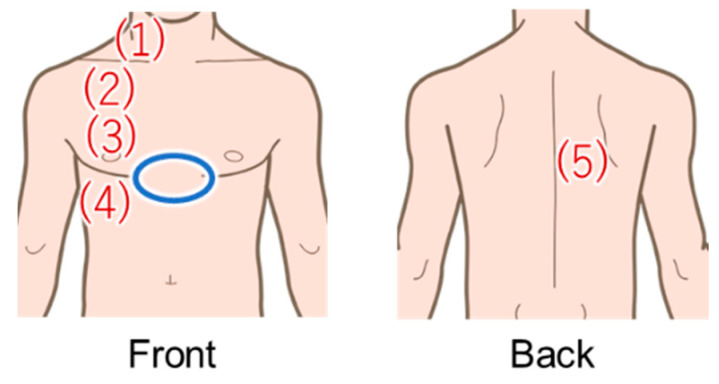
Areas in red were measured using the proposed method, and areas in blue were measured with the Go Direct Respiration Belt for the reference value.

**Figure 8 sensors-24-01626-f008:**
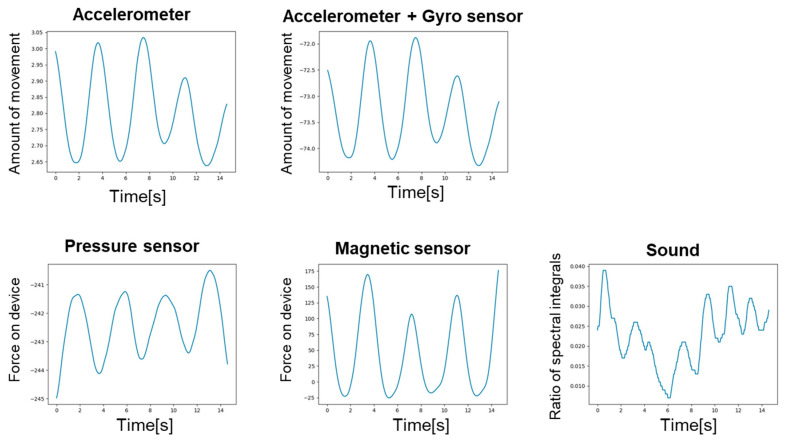
Breathing waveform for each measurement method.

**Figure 9 sensors-24-01626-f009:**
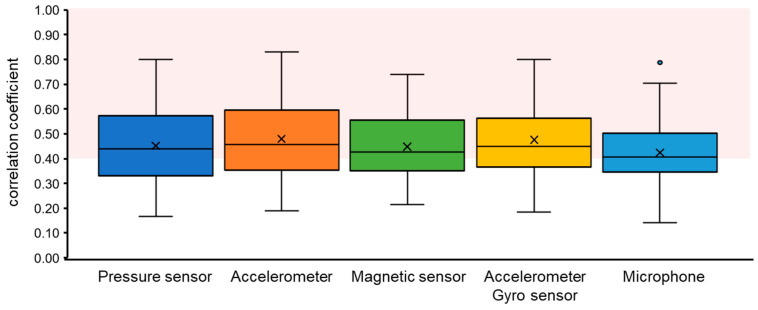
Comparison of breathing waveform acquisition accuracies by measurement method. The area shown in red color indicates where the correlations are considered to exist (>0.4).

**Table 1 sensors-24-01626-t001:** Pressure sensor specifications.

Thickness	0.20–1.25 mm
Pressure sensitive range	0.2–10.0 N
Minimum sensitivity	20–100 g

**Table 2 sensors-24-01626-t002:** Specifications of the 9-axis inertial sensor.

FXOS8700 3-axis accelerometer	Resolution	14 bit
	Detection range	−2 to 2 g
FXOS8700 3-axis magnetic Sensor	Resolution	16 bit
	Detection range	−1200 to 1200 µT
FXAS21002 3-axis gyro sensor	Resolution	16 bit
	Detection range	−250 to 250 degree/s

**Table 3 sensors-24-01626-t003:** Microphone specifications.

Directivity	Omnidirectional
Frequency response	50–15,000 Hz
Frontal sensitivity	−38 ± 3.5 dB

**Table 4 sensors-24-01626-t004:** Go Direct Respiration Belt specifications.

Detection range	0–50 N
Resolution	0.01 N
Sampling rate	20 Hz

**Table 5 sensors-24-01626-t005:** Analysis of variance table within three-factor participants (A: breathing method, B: measurement point, C: measurement method, SS: sum of squares, Df: degrees of freedom, MS: mean square), ns: not significant, * *p* < 0.05, ** *p* < 0.01.

Variable Name	SS	Df	MS	F Value	*p*-Value		$\mathrm{Partial}\;\eta^{2}$
A	0.968	1.00	0.968	27.703	0.000	**	0.681
B	0.874	4.00	0.219	6.218	0.000	**	0.324
C	0.648	2.13	0.304	7.066	0.003	**	0.352
A × B	0.012	4.00	0.003	0.072	0.990	ns	0.006
A × C	0.485	2.53	0.192	8.471	0.000	**	0.395
B × C	0.545	16.00	0.034	2.012	0.014	*	0.134
A × B × C	0.262	16.00	0.016	1.146	0.315	ns	0.081

**Table 6 sensors-24-01626-t006:** Mean values and standard deviations of correlation coefficients for the breathing method and measurement method factors under each condition (A: breathing method, A1: normal breath, A2: deep breath, C: measurement method, C1: pressure sensor, C2: accelerometer, C3: magnetic sensor, C4: accelerometer + gyro sensor, C5: microphone).

Condition A	Condition C	Number of Data	Mean Value	SD
A1	C1	70	0.485	0.164
A1	C2	70	0.396	0.159
A1	C3	70	0.437	0.141
A1	C4	70	0.391	0.156
A1	C5	70	0.443	0.142
A2	C1	70	0.533	0.154
A2	C2	70	0.501	0.179
A2	C3	70	0.562	0.156
A2	C4	70	0.502	0.179
A2	C5	70	0.426	0.113

**Table 7 sensors-24-01626-t007:** Mean values and standard deviations of correlation coefficients for the position and method factors under each condition (B: measurement point, C: measurement method, C1: pressure sensor, C2: accelerometer, C3: magnetic sensor, C4: accelerometer + gyro sensor, C5: microphone).

Condition B	Condition C	Number of Data	Mean Value	SD
B1	C1	28	0.450	0.128
B1	C2	28	0.411	0.192
B1	C3	28	0.405	0.154
B1	C4	28	0.406	0.187
B1	C5	28	0.453	0.141
B2	C1	28	0.511	0.176
B2	C2	28	0.524	0.193
B2	C3	28	0.549	0.152
B2	C4	28	0.515	0.195
B2	C5	28	0.433	0.151
B3	C1	28	0.502	0.177
B3	C2	28	0.389	0.162
B3	C3	28	0.468	0.138
B3	C4	28	0.394	0.169
B3	C5	28	0.427	0.126
B4	C1	28	0.559	0.150
B4	C2	28	0.481	0.170
B4	C3	28	0.590	0.160
B4	C4	28	0.492	0.170
B4	C5	28	0.434	0.092
B5	C1	28	0.523	0.157
B5	C2	28	0.438	0.139
B5	C3	28	0.485	0.142
B5	C4	28	0.425	0.134
B5	C5	28	0.425	0.132

**Table 8 sensors-24-01626-t008:** Test results of the simple main effects for the interactions between the breathing and measurement method factors as well as for interactions between the measurement position and measurement method factors. (A: breathing method, A1: normal breath, A2: deep breath, B: measurement point, C: measurement method, C1: pressure sensor, C2: accelerometer, C3: magnetic sensor, C4: accelerometer + gyro sensor, C5: microphone, SS: sum of squares, Df: degrees of freedom, MS: mean square) ns: not significant, * *p* < 0.05 ** *p* < 0.01.

Slice	Variable Name	SS	Df	MS	F Value	*p*-Value		$\mathrm{Partial}\;\eta^{2}$
C = C1	A	0.081	1.000	0.081	4.032	0.066	ns	0.237
C = C2	A	0.388	1.000	0.388	23.340	0.000	**	0.642
C = C3	A	0.546	1.000	0.546	34.927	0.000	**	0.729
C = C4	A	0.427	1.000	0.427	23.861	0.000	**	0.647
C = C5	A	0.010	1.000	0.010	0.463	0.508	ns	0.034
A = A1	C	0.545	1.970	0.209	5.420	0.011	*	0.294
A = A2	C	0.262	2.690	0.268	9.886	0.000	**	0.432
C = C1	B	0.176	4.000	0.044	1.833	0.136	ns	0.124
C = C2	B	0.326	4.000	0.082	3.680	0.010	*	0.221
C = C3	B	0.582	4.000	0.145	7.297	0.000	**	0.360
C = C4	B	0.321	4.000	0.080	3.869	0.008	**	0.230
C = C5	B	0.014	4.000	0.004	0.222	0.925	ns	0.017
B = B1	C	0.066	1.960	0.034	0.753	0.479	ns	0.055
B = B2	C	0.213	2.640	0.081	2.245	0.108	ns	0.147
B = B3	C	0.261	2.160	0.121	3.846	0.031	*	0.228
B = B4	C	0.441	1.540	0.287	7.600	0.006	**	0.369
B = B5	C	0.211	1.960	0.107	3.943	0.033	*	0.233

**Table 9 sensors-24-01626-t009:** Results of multiple comparisons (Bonferroni method) (A: breathing method, A1: normal breath, A2: deep breath, B: measurement position, C: measurement method, C1: pressure sensor, C2: accelerometer, C3: magnetic sensor, C4: accelerometer + gyro sensor, C5: microphone, Df: degrees of freedom).

Level Pair	Terms	Diff	t-Value	Df	*p*-Value	ADJUSTMENT *p*-Value	Order
C1–C4	A1	0.093	3.555	13.000	0.004	0.035	C1 > C4
C1–C2	A1	0.089	3.432	13.000	0.005	0.035	C1 > C2
C3–C5	A2	0.136	5.542	13.000	0.000	0.001	C3 > C5
C1–C5	A2	0.107	4.519	13.000	0.001	0.004	C1 > C5
B1–B4	C3	−0.185	5.088	13.000	0.000	0.002	B1 < B4
B1–B2	C3	−0.144	4.053	13.000	0.001	0.008	B1 < B2
B3–B4	C3	−0.122	3.143	13.000	0.008	0.047	B3 < B4
B2–B5	C4	0.089	3.542	13.000	0.004	0.036	B2 > B5
B1–B2	C4	−0.108	3.313	13.000	0.006	0.036	B1 < B2
C3–C5	B4	0.156	6.893	13.000	0.000	0.000	C3 > C5
C1–C5	B4	0.125	5.144	13.000	0.000	0.001	C1 > C5

**Table 10 sensors-24-01626-t010:** Average correlation coefficients by method selection.

Before method selection	0.46 ± 0.14
After method selection	0.52 ± 0.14

**Table 11 sensors-24-01626-t011:** Breathing waveform calculations by changing the frequency band Mean value of correlation coefficient.

Measurement point (1)	300–600 Hz	0.40 ± 0.09
	500–800 Hz	0.43 ± 0.10
Measurement point (2)	300–600 Hz	0.42 ± 0.09
	500–800 Hz	0.48 ± 0.14
Measurement point (3)	300–600 Hz	0.44 ± 0.15
	500–800 Hz	0.45 ± 0.16
Measurement point (4)	300–600 Hz	0.47 ± 0.17
	500–800 Hz	0.43 ± 0.12
Measurement point (5)	300–600 Hz	0.42 ± 0.13
	500–800 Hz	0.47 ± 0.13

## Data Availability

Data are contained within the article.

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
