# Peer review of "Breath Measurement Method for Synchronized Reproduction of Biological Tones in an Augmented Reality Auscultation Training System"

_sensors, 2024, doi:10.3390/s24051626_

Round 1
Reviewer 1 Report
Comments and Suggestions for Authors
I find the paper interesting and worthwhile research to do.
However, the paper can, and should, be improved:
1) Page 1 line 20, why is one correspondence email hyperlinked and the other not?
2) It is somewhat difficult to read the paper because the reference pointers, e.g. [1], have been placed at the end of the sentence, not after the author's name.
3) A lot of the technology used in the paper is very generic, off the shelf, even hobby devices, it does not create a state of the art presentation. I would recommend for the authors to focus the paper on their research contribution, rather than a description of how they connected some off the shelf components in the way they are intended to be connected. After all, the technology used is not state of the art anyway, and there are many better ways of implementing what the authors propose.
4) If figure 8 required? It is again an off the shelf device that is not part of this research.
5) The paper states ": Informed consent was obtained from all subjects involved in the study.", but no statement on the ethics approval, who approved and and ethics approval number number is given.
6) The reference list is not presented correctly. Clearly from a different publisher!
7) This paper presents 19 references, there is no ideal number of references, but 19 references these days seems far far too little to present a clear understanding of the state of the art.
Comments on the Quality of English LanguageNot bad.
Reviewer 2 Report
Comments and Suggestions for Authors
Dear Authors,
Congratulations on the clear and well-written review manuscript. These are some points for your consideration to revise. Kudos to your team.
Check the correspondences, check (H.K.) and the rest in brackets. Are they necessary?
In line 47, possible to add more references?
In lines 97-104, highlight the novelty of your manuscript as well. Your abstract was well-written. Good job.
Are there any potential limitations of the device design? Do highlight if there is any.
In the experiments, briefly elaborate on how deep and normal breath affect the results.
Why Go Direct Respiration Belt was specifically chosen for the reference?
In the conclusion, state how will this manuscript impact the field.
Briefly elaborate on why the correlation coefficient is significant.
Good luck!
Reviewer 3 Report
Comments and Suggestions for Authors
1. Does the stethoscope developed by the authors come into contact with the chest wall through handheld means? How to overcome the artifacts caused by hand tremors and friction with the chest wall of the operator? How to identify and eliminate such artifacts? Can it be completely eliminated by low-pass filtering?
2. In section 3.1,five measurement points (four in the front and one in the back) were selected from the basic auscultation positions for the breath sounds (Figure 7).
The respiratory sound collection location was only selected on the right side, why not on the both side at the same time?
How to standardize the collection location for subjects with different body types?
3. In section 3.3,the average value of the correlation coefficient for each measurement method was more than 0.4, which is considered to be correlated.
The correlation coefficient r (approximately 0.4) is relatively low and needs to be further improved through equipment and algorithms.
4. During respiratory disease attacks or mechanical ventilation, respiratory sounds become very complex. Hardware devices, sampling methods, and algorithms need to be tested and validated in more healthy subjects and real patients with different respiratory diseases.
Comments on the Quality of English Languageno
Reviewer 4 Report
Comments and Suggestions for Authors
The article is well written, quite detailed. I would like to to highlight an interesting point, as the influence of the measurement location and the way of breathing on the quality of registration depending on the type of sensor chosen, that for better accuracy it is better to use several breathing sensors at once.
The authors investigated the correlation, 0.5 correlation is small, but the breath signal is such a noisy signal that it is already a relatively good indicator. Have the authors evaluated the required minimum value of the correlation coefficient?
The authors write that they make a simulator, as I understand for training doctors, but they study only breathing of healthy people, I think for doctors are important and cases of different pathologies, how the signal changes in them, there are different noises in pathologies and these noises and the useful component of the signal overlap quite strongly in the frequency range. This is not representative enough if we work only with signals from healthy patients.
The authors do not describe whether the noise will be studied further, they only write about improving the correlation in the future.
I highly recommend improving the introduction by reviewing more relevant references. I also suggest fixing the abstract by adding more specific details and quantitative study results
Round 2
Reviewer 1 Report
Comments and Suggestions for Authors
Looking at the past review comments:
1) Page 1 line 20, why is one correspondence email hyperlinked and the other not?
The authors still have it wrong. The template has no hyperlinks.
6) The reference list is not presented correctly. Clearly from a different publisher!
It is stil not correct.
Ref [1] "Respir" should have a full stop.
Ref [2] "Ann" should have a full stop.
Ref [3] "Arch" should have a full stop.
Ref [5] "Clin" and "Pediatr" should have a full stop. Text "(Phila)." should be deleted.
Ref [6] " Global Pediatric Health" should be presented in shortened form.
Ref [10] "Med" should have a full stop.
Ref [11] " Teaching and learning in medicine" should be capitalized and presented in shortened form.
etc.
I am still not sure about the level of off the shelf components discussion that is being made here. It is not of the high level of archival research that an archival journal should incorporate.
Comments on the Quality of English LanguageN/A
Reviewer 4 Report
Comments and Suggestions for Authors
Thanks for the detailed answers to the questions. All necessary changes and additions are done.
Author Response
Dear reviewer:
Thank you for your sincere comments.
These have greatly helped us to improve the quality of our manuscript.
Best regards,
Yukiko Kono and Toshiya Nakaguchi